# Adenosine/β-Cyclodextrin-Based Metal–Organic Frameworks as a Potential Material for Cancer Therapy

**DOI:** 10.3390/biom13071154

**Published:** 2023-07-20

**Authors:** Rajaram Rajamohan, Sekar Ashokkumar, Mani Murali Krishnan, Kuppusamy Murugavel, Moorthiraman Murugan, Yong Rok Lee

**Affiliations:** 1School of Chemical Engineering, Yeungnam University, Gyeongsan 38541, Republic of Korea; 2Plasma Bioscience Research Center, Department of Electrical and Biological Physics, Kwangwoon University, Seoul 01897, Republic of Korea; kumarebt@gmail.com; 3Department of Chemistry, Bannari Amman Institute of Technology, Sathyamangalam 638 401, Tamil Nadu, India; muralikrishnan@bitsathy.ac.in; 4PG and Research Department of Chemistry, Government Arts College, Chidambaram 608 102, Tamil Nadu, India; ksmvel@gmail.com; 5Department of Chemistry, IFET College of Engineering, Villupuram 605 108, Tamil Nadu, India; m_murugan79@rediffmail.com

**Keywords:** adenosine, β-cyclodextrin, inclusion complex, metal–organic framework, cancer cells

## Abstract

Recently, researchers have employed metal–organic frameworks (MOFs) for loading pharmaceutically important substances. MOFs are a novel class of porous class of materials formed by the self-assembly of organic ligands and metal ions, creating a network structure. The current investigation effectively achieves the loading of adenosine (ADN) into a metal–organic framework based on cyclodextrin (CD) using a solvent diffusion method. The composite material, referred to as ADN:β-CD-K MOFs, is created by loading ADN into beta-cyclodextrin (β-CD) with the addition of K^+^ salts. This study delves into the detailed examination of the interaction between ADN and β-CD in the form of MOFs. The focus is primarily on investigating the hydrogen bonding interaction and energy parameters through the aid of semi-empirical quantum mechanical computations. The analysis of peaks that are associated with the ADN-loaded ICs (inclusion complexes) within the MOFs indicates that ADN becomes incorporated into a partially amorphous state. Observations from SEM images reveal well-defined crystalline structures within the MOFs. Interestingly, when ADN is absent from the MOFs, smaller and irregularly shaped crystals are formed. This could potentially be attributed to the MOF manufacturing process. Furthermore, this study explores the additional cross-linking of β-CD with K through the coupling of -OH on the β-CD-K MOFs. The findings corroborate the results obtained from FT-IR analysis, suggesting that β-CD plays a crucial role as a seed in the creation of β-CD-K MOFs. Furthermore, the cytotoxicity of the MOFs is assessed *in vitro* using MDA-MB-231 cells (human breast cancer cells).

## 1. Introduction

Metal–organic frameworks (MOFs) primarily consist of metal ions and organic ligands [1,2]. The manipulation of the arrangement processes of different materials is crucial for modifying their properties, which can involve altering their shape, dimension, aggregation, or composition [3,4,5,6]. Through coordination chemistry, MOFs enable the attainment of diverse structural dimensions within the material. Multimodal ligands are employed to connect metal ions and form secondary building units in the MOF structure. MOFs based on supramolecular-cyclodextrin (β-CD-MOFs) have attracted a lot of interest lately because of their distinctive characteristics and prospective uses [7,8,9]. CD-MOFs combine the host–guest properties of CDs with the structural versatility of metal–organic frameworks, resulting in a highly functional and tunable material system. CD-MOFs are formed by the assembly of β-CD units with metal ions or clusters, creating a porous and crystalline framework [10]. The inclusion of β-CD units provides a cavity-like structure within the MOF, allowing for guest molecule encapsulation and controlled release [11,12]. This feature opens up possibilities for various applications, including drug delivery, gas storage, sensing, and catalysis [13,14,15,16,17,18,19]. The utilization of biomolecules as organic ligands holds great promise due to their biocompatibility. Biomolecules possess a variety of coordination modes, thereby offering a wide range of metal-binding sites [20].

In particular, MOFs incorporating CD have emerged as a promising approach for cancer therapy, offering enhanced drug delivery and targeting capabilities. The combination of these two components has paved the way for novel strategies in cancer treatment. By incorporating CD into MOFs, researchers have achieved efficient encapsulation and controlled release of therapeutic agents, addressing the challenges associated with conventional drug delivery systems [21]. The porous structure of MOFs allows for high drug-loading capacities, ensuring optimal delivery of anticancer drugs to tumor sites [22]. Furthermore, the inclusion of CDs enhances the stability and solubility of encapsulated drugs, improving their bioavailability [23]. MOFs with CD can be functionalized with targeting ligands, enabling specific recognition of cancer cell markers and selective accumulation at the tumor site [24]. This targeted approach minimizes off-target effects and enhances the therapeutic efficacy of anticancer drugs [25]. Additionally, the tunable properties of MOFs, such as pore size and surface chemistry, provide control over drug release kinetics, enabling sustained and controlled drug delivery [26]. The combination of MOFs and CDs has shown great potential in advancing cancer therapy and personalized medicine. Continued research and development in this field aim to optimize drug delivery systems, improve treatment outcomes, and reduce side effects [27].

Solvothermal methods are commonly employed for the synthesis of MOFs, which involve heating the metal ions and ligands within a solvent [28]. Other techniques, including microwave-assisted, electrochemical, ball milling, and camera flash, have also been utilized for the synthesis [29,30,31]. However, MOFs synthesized using these routes are non-recyclable due to the presence of unsafe components, such as metal ions and ligands, which limit their potential for biological applications [32]. To address these concerns, it is advisable to explore alternative synthetic routes that operate at room temperature. One promising approach involves the use of biomolecule ligands (such as cyclodextrins, peptides, and amino acids) and biocompatible metal ions (such as calcium, potassium, and sodium), which can mitigate the risks associated with MOFs [20].

The aim of developing cyclodextrin-based MOFs in cancer therapy is to enhance the host–guest encapsulation capability of the material. Incorporating CDs into MOFs can lead us to improve the performance in applications such as drug delivery, where the controlled release of encapsulated drugs is desired. Also, they allow for the development of advanced drug delivery systems that improve the selectivity, efficacy, and safety of cancer treatments. These innovative platforms have the potential to revolutionize cancer therapy by overcoming drug resistance, enabling targeted therapy, and providing personalized treatment options for better patient outcomes.

## 2. Materials and Methods

### 2.1. Materials

β-Cyclodextrin (β-CD, C_42_H_70_O_35_, ≥99.0% purity, and molecular weight: 1134.98), potassium phosphate (KH_2_PO_4_, ≥99.0% purity, and molecular weight: 136.09), adenosine (ADN, C_10_H_13_N_5_O_4_, >99.0% purity, and molecular weight: 267.24), and methanol (MeOH) are of analytical grade and bought from Sigma-Aldrich Co., Ltd., Seoul, Republic of Korea. In our lab, deionized water is prepared.

### 2.2. Synthesis of β-CD-K MOFs

To synthesize β-CD-K MOFs, an improved methanol (solvent) vapor diffusion method is employed. Detailed information for the synthesis of MOFs can be found in the Appendix A.

### 2.3. ADN Loading to β-CD-K MOFs (ADN:β-CD-K MOFs)

In this study, the focus was on investigating the adsorption of ADN onto β-CD-K MOFs using impregnation, as illustrated in Figure 1. To initiate the process, a non-homogeneous solution was created by combining 40 mg of obtained MOFs with a 4 mg/mL concentration of ADN in a 10 mL ethanol solution. The resulting solution was then agitated at 400 rpm for a duration of 48 h. To separate the ADN-loaded inclusion complexes (ICs), centrifugation was employed, and the solid obtained was subjected to multiple ethanol washes. Subsequently, the solid was dried under a vacuum at a temperature of 40 °C overnight. The resulting product derived from this process was named as ADN:β-CD-K MOFs.

### 2.4. Details of Quantum Mechanical Calculations

The semi-empirical quantum mechanical computations were conducted using the PM3 method in Gaussian 16.

### 2.5. In-Vitro Release Profile of ADN from MOFs

The detailed experimental procedure for the *in vitro* release profile can be accessed in the Appendix A.

### 2.6. In-Vitro Cytotoxic Assay

The detailed experimental procedure for the *in-vitro* cytotoxic assay can be found in the Appendix A.

### 2.7. Characterization of ADN:β-CD-K MOFs

The synthesized MOFs underwent characterization using various analytical techniques, including XRD, DSC, FE-SEM, AFM, XPS, FT-IR, Raman spectroscopy, and ^1^H NMR. Detailed information about the instruments and methodologies employed for each technique can be found in the Appendix A.

## 3. Results and Discussion

### 3.1. Examination of the Interaction of ADN and β-CD-K MOFs in the Virtual State

Before characterizing ADN:β-CD-K MOFs, it is essential to examine the interaction between ADN and β-CD in the form of MOFs, focusing on the hydrogen bonding interaction, and energy parameter in greater detail.

#### 3.1.1. Energetically Favorable Orientation via Single-Point Energy Computation

The determination of ADN’s orientation towards the host in the virtual state for inclusion complexes (ICs) is achieved through single-point energy calculations [33]. In the ICs, there are two distinct orientations, represented as orientation 1 and orientation 2. For determining the single-point energy of orientation 1, where the benzene ring section of ADN is oriented towards the 2° rim of β-CD (referred to as ICs of orientation 1, as shown in Figure 1A,C), the PM3 method is utilized. Similarly, for orientation 2, which involves the quinone ring part of ADN oriented towards the 2° rim of β-CD (referred to as ICs of orientation 2, depicted in Figure 1B,D), the single-point energy is computed.

The calculation of the energy for the ICs (ΔE) between ADN and β-CD is performed using a semi-empirical calculation method on the optimized energy structure. The following equation is utilized for this purpose.
ΔE = E_β-CD/ADN_ − (E_β-CD_ + E_ADN_)

The E_β-CD_, E_ADN_, and E_β-CD/ADN_ represent the optimized energy of free β-CD, ADN, and ICs of ADN with β-CD, respectively. Table 1 presents the consolidation of a thermodynamically favorable inclusion process in a vacuum, as indicated by the negative energy during the formation of ICs. The complexation energy for orientation 1 and orientation 2 is found to be −105.835 KJ/mol and −92.445 KJ/mol, respectively. Based on the complexation energy, it can be concluded that orientation 1 inclusion is thermodynamically favorable. Figure 2 depicts the hydrogen bonding interactions between different atoms in both orientations. It provides a visual representation of these interactions. Additionally, Table 2 presents the comprehensive results related to these hydrogen bonding interactions, offering detailed information and data for further analysis.

#### 3.1.2. ΔE between HOMO and LUMO of Frontier Molecular Orbitals of ICs

The stability of ICs can be assessed by utilizing the HOMO and LUMO energy levels [34]. The stability of the molecules can be understood by examining the energy differences between the energy levels (E_HOMO_-E_LUMO_). Among the various orientations of ICs-MOFs, orientation 2 stands out as a stabilized model and is highly favorable due to its higher E_HOMO_-E_LUMO_ value compared to ICs with orientation 1 (as shown in Table 3 and Figure 3). Generally, ICs with higher values of E_HOMO_-E_LUMO_ exhibit greater stability [35,36].

### 3.2. Spectral Analysis of ADN:β-CD-K MOFs

#### 3.2.1. XRD Pattern Analysis

Figure 4A presents the XRD results, which provide evidence of the exceptional crystalline nature of β-CD-K MOFs [37,38]. The β-CD-K MOFs exhibited a prominent peak at 8.87°, 12.81°, and 18.91° [39,40,41]. Comparing the experimental diffraction patterns with the simulated peaks demonstrates an excellent agreement, indicating that the synthesized material is present in the pure phases. Nevertheless, the differences observed in peak intensities between the experimental patterns and simulated values suggest that the variation in sample orientations could be responsible for this disparity. For ADN, distinct major peaks are observed at 11.43°, 15.14°, 17.42°, 22.85°, 24.41°, 27.36°, 30.57°, and 31.83°, which indicate the crystalline nature of ADN [42]. When ADN is loaded into β-CD-K MOFs, the predominant peaks exhibit slight shifts in their positions, and some peaks merge with those of β-CD-K MOFs. Additionally, some peaks have reduced intensity but are still noticeable. The observed modifications in the peaks related to the ADN-loaded ICs within the MOFs indicate the incorporation of ADN into a partially amorphous state.

#### 3.2.2. DSC Analysis

The β-CD-K MOFs analyzed through DSC exhibited an absence of water/moisture content, as evidenced by the absence of endothermic peaks within the temperature range of 60–110 °C (Figure 4B). However, a notable broad endothermic peak is observed at 188.92 °C, indicating a melting process with the heat of melting (ΔH_melting_) which is about 860.9 J/g. This peak signifies a transformation of the crystalline state of β-CD [43]. This study also established a correlation between the DSC behavior of β-CD-K MOFs and the interaction of potassium (K) within the MOFs [44]. Conversely, the DSC analysis of ADN:β-CD-K MOFs displayed a broad endothermic peak at 108 °C, accompanied by heat of melting of approximately 138.2 J/g. This peak represents the release of water/moisture content from the ADN:β-CD-K MOFs. Interestingly, no endothermic peaks are observed in the temperature range of 180–190 °C, indicating that the obtained MOFs possess a semi-crystalline nature. These findings align well with the obtained XRD results. Additionally, a distinct small endothermic change is observed around 226.34 °C in the DSC analysis, accompanied by heat of melting of 1394 J/g. This change is attributed to the transformation of ADN:β-CD within the MOFs, without any weight loss. Furthermore, the DSC outputs indicated that ADN is uniformly distributed in the MOFs, as evidenced by the endothermic change at the same temperature of 226.34 °C and heat of 1394 J/g [45]. This distribution could be influenced by the degradation of the MOFs.

#### 3.2.3. FE-SEM Analysis

Surface morphological visualization of the prepared MOFs is examined through FE-SEM analysis, enabling clear visual representations [46,47]. The SEM images revealed well-defined crystalline structures of the MOFs (Figure 5). Interestingly, the absence of ADN in the MOFs led to the formation of smaller and irregularly shaped crystals, potentially resulting from the MOF manufacturing process. However, upon loading with ADN, the original small crystalline appearance is completely transformed, and larger crystals are observed. To gain insights into the elemental composition of the obtained MOFs, energy dispersive X-ray spectroscopy (EDX) and elemental mapping are measured. Figure 5C,H display the elemental maps, indicating the presence of carbon (C), oxygen (O), and potassium (K) within the MOFs. In addition, the ratios of these elements are also provided, and it is worth noting that the atomic weight percentages of carbon (C) and oxygen (O) did not exhibit significant changes in the MOFs before and after ADN loading. Additionally, the elemental analysis confirmed the presence of individual C, O, and K atoms within the formed MOFs, as evidenced by the elemental mapping (Figure 5D–G,I–M).

#### 3.2.4. AFM Analysis

The AFM analyses of both MOFs are depicted in Figure 6. The height or topology images clearly differentiate each MOF particle based on color contrast [48]. Similar color contrast is seen in the results of surface potential measurement on the topology scan, and this pattern is seen in the line scan analysis for both the surface potential and the height of the MOFs. Additionally, the MOF particles display a surface roughness on the scale of tens of nanometers. Both MOFs exhibit negligible variations in morphology and surface roughness. The ADN-loaded MOFs exhibit a surface with uniformly distributed roughness comparable to that of the ADN-free MOFs.

#### 3.2.5. XPS Analysis

The analysis presented in Figure 7 and Appendix A provides insights into the assumed binding of the β-CD-MOFs and ADN:β-CD-K MOFs. XPS measurements offer supporting evidence for the interaction of β-CD, and also ADN:β-CD-ICs with MOFs. The survey spectrum of β-CD-K MOFs shows a peak indicating the elements of C 1s, O 1s, and K 2p in the MOFs. The binding energy (BE) of the C 1s peaks remains consistent at 285.68 eV for both MOFs. The atomic percentage composition of ADN:β-CD-K MOFs is found to be nearly identical to that of the MOFs without ADN loadings. The deconvolution of peaks centered at 284 and 286 eV, representing the bonds of C-C and C-O, respectively, suggests the presence of excess CDs (unreacted CDs) on ADN [49]. While the atomic ratios show no noticeable changes, the BE of the fitted O 1s peak increases (the measured BE is at 531.98 eV for the material, β-CD-K MOFs). This increase in BE is attributed to the higher 1s atomic ratio of ADN:β-CD-K MOFs related to that of K 2p, resulting in a decrease in the electron density values surrounding the O element and an increase in the BE of O-K with increasing growing time [50,51]. The K 2p_3/2_ peak is detected at 292.18 eV, and the K 2p_1/2_ peak is detected at 295.18 eV in the high-resolution spectra of K 2p, indicating the presence of K^+^ in the MOFs [49]. The ADN:β-CD-K MOFs synthesized for 80 h exhibit the lowest K 2p atomic ratio of 0.85% among the β-CD-K MOFs, suggesting the formation of dendricolloids due to prolonged growth. This can be attributed to the additional cross-linking of β-CD with K through the coupling of hydroxyl groups on the β-CD-K MOFs. This finding aligns with the results of FT-IR spectral analysis and indicates that β-CD acts as a seed in the formation of MOFs [52,53]. Though, after the loading of ADN, the BEs and relative contents undergo slight changes, suggesting that interactions occur during the adsorption process of ADN.

#### 3.2.6. FT-IR Spectral Interpretation

In Figure 8A, the FT-IR spectrum of ADN exhibited several prominent peaks [42]. The peak observed at 3345.21 cm^−1^ is assigned to the stretching vibration of −NH_2_, while the peak at 3166.34 cm^−1^ is attributed to the stretching vibration of −OH. Additionally, other peaks are observed at the following wavenumbers: 2638.31 cm^−1^ (stretching vibration of −CH_2_), 1675.12 cm^−1^ (stretching vibration of aromatic C–C bonds), 1606.11 cm^−1^ (stretching vibration of aromatic C=C bonds), 1341.16 cm^−1^ (associated with the stretching vibration of aromatic tertiary amine C–N bonds), 1295.55 cm^−1^ (stretching vibration of aromatic primary amine C–N bonds), 1202.79 cm^−1^ (stretching vibration of aromatic C–H bonds), and 1043.24 cm^−1^ (stretching vibration of aromatic C–O–C bonds). In Figure 8, the FT-IR spectrum of β-CD displayed distinctive features [44,54]. A broadband with a transmittance peak at 3388.55 cm^−1^ indicated the symmetric and asymmetric stretching vibrations of O–H bonds, arising from the intermolecular hydrogen bonding of β-CD. Furthermore, a sharp absorption band is observed at 2928.64 cm^−1^, which corresponded to the stretching vibration of C–H bonds. The absorption band at 1651.14 cm^−1^ resulted from the bending of H–O–H bonds. Additionally, absorption bands appeared at 1160.53 cm^−1^ (asymmetric stretching vibration of C–O–C bonds) and 1028.79 cm^−1^ (symmetric stretching vibration of C–O–C bonds). In the FT-IR spectrum, the β-CD-K MOFs reveal peaks that are slightly shifted in position, confirming the existence of β-CD within the synthesized MOFs. These observed shifts in the bands suggested a strong interaction of β-CD with K^+^ ions in the MOFs. As demonstrated upon the comparing of the FT-IR spectra of β-CD and MOFs, the overall structure of β-CD is maintained in the MOFs. In MOFs without ADN, the IR signal seems to overlap with that of β-CD. Consequently, upon the formation of MOFs with K ions, there are no noticeable shifts or disappearances of peaks [44].

However, MOFs containing ADN show slight variations in stretching frequency values. It is noteworthy that the stretching frequency peak for β-CD appears at 1646 cm^−1^, primarily due to the presence of water molecules within the β-CD cavity. Furthermore, the MOFs retain their ability to encapsulate water molecules within their cavities. When ADN is introduced into the MOFs, the peak diminishes significantly, suggesting that ADN replaces the water molecules, forming inclusion complexes (ICs) (shoulder peak). The FT-IR spectrum of ADN:β-CD-K MOFs reveals the presence of distinct ADN bands, which serve as evidence for the loading of ADN into β-CD-K MOFs. However, the minimum weight percentage of ADN in the MOFs makes it challenging to distinguish certain peaks clearly. Nevertheless, these adjustments provide additional confirmation of the effective encapsulation of ADN by β-CD-K MOFs.

#### 3.2.7. Raman Spectral Interpretation

Figure 8B illustrates the Raman spectra of the three substances: ADN, β-CD-K MOFs, and ADN:β-CD-K MOFs. The Raman spectra of ADN exhibit two significant stretching vibrations, as discussed below. The peak observed at 725 cm^−1^ corresponds to the adenine ring present in the guest molecule, while the peak at 1342 cm^−1^ is attributed to the NH_2_ vibration [55,56,57]. In the case of β-CD-MOF-K MOFs, their Raman spectra display a peak at 3027 cm^−1^, which can be assigned to the C–H stretching mode [58,59]. Additionally, a broad peak appears at 3328 cm^−1^, indicating the O–H stretching mode. The Raman spectra of the ADN-loaded MOFs (ADN:β-CD-K MOFs) do not display distinct peaks and indicate a significantly lower quantity of ADN loaded into the MOFs. The primary peaks merge with those of the β-CD-K MOFs.

#### 3.2.8. ^1^H NMR Spectral Interpretation

##### ^1^H NMR Spectral Interpretation of β-CD-K MOFs

The ^1^H NMR spectrum revealed several proton signals corresponding to different regions of the β-CD ring. The methine and methylene protons are observed as multiplets at 3.30, 3.56, and 3.63 ppm, indicating their presence in the spectrum. In addition, a triplet peak at 4.45 ppm, exhibiting a coupling constant of about 12 Hz, is identified as the methine proton belonging to the hydroxy methylene group connected to the carbon (refer to Figure 9A). Two signals appeared at 4.82 and 5.67 ppm, presenting as doublets with a small coupling constant. These signals are attributed to the hydroxy groups occupying the equatorial position within the cyclodextrin ring. Moreover, an equatorial hydrogen signal at 5.73 ppm, accompanied by a coupling constant of approximately 6.6 Hz, is assigned to the proton located on the β-CD molecule. In addition, a multiplet at 3.33 ppm is conveniently assigned to the hydroxy proton attached to the methylene group. For a comprehensive overview of the proton chemical shifts of β-CD, please refer to Appendix A.

##### ^1^H NMR Spectral Interpretation of ADN

The ^1^H NMR spectrum of the ADN molecule reveals specific proton signals that offer valuable insights into its structural composition. The aromatic protons appeared as singlets at 8.14 and 8.35 ppm, representing H-8 and H-2, respectively. The singlet at 7.36 ppm, with a broad peak and two proton integral values, is assigned to the amino protons at C-6 (Figure 9C). The proton signals arising from the ribose molecule are detected within the range of 3.56 to 5.88 ppm. The hydroxy protons at C-2′ and C-3′ are observed as doublets at 5.46 and 5.25 ppm, respectively, with a coupling constant of 4 Hz. Additionally, the hydroxy proton attached to the methylene carbon appeared as a multiplet at 3.97 ppm. The methine proton at C-1′ is assigned to the doublet observed at 5.88 ppm, with a coupling constant of 4 Hz. Furthermore, five multiplets within the range of 3.56–5.47 ppm are attributed to the methine protons at C-2′, C-3′, and C-4′, and the methylene protons at C-5′. These assignments are developed based on the electronic environments of the compounds. A downfield multiplet is observed at 5.43–5.47 ppm, representing the methine proton at C-4′ with one proton integral. Two multiplets observed in the upfield region are assigned to the methylene protons at C-5′. Therefore, the multiplets at 3.54–3.58 and 3.66–3.69 ppm are conveniently assigned to the methylene protons. The remaining two multiplets, observed at 4.14–4.16 and 4.60–4.63 ppm, are attributed to the methine protons of C-3′ and C-2′, respectively [60]. For a detailed overview of the proton chemical shifts of ADN, please refer to Appendix A.

##### ^1^H NMR Spectral Interpretation of ADN:β-CD-K MOFs

The confirmation of MOF formation is achieved by analyzing the NMR spectral pattern of the ICs. The NMR spectrum of the ICs provides all the proton signals necessary for characterization. Confirmation is obtained by analyzing the proton integral values of ADN in the ADN-β-CD-ICs, indicating the encapsulation of a single ADN molecule with β-CD, and the spectrum is provided in Appendix A [42]. The analysis of the ROESY provided further confirmation of the encapsulation. Appendix A shows the ROESY for the ICs of ADN and β-CD. Specifically, the proton signal at 5.85 ppm displayed correlations with signals at 3.32, 4.61, 5.45, 5.70, and 8.35 ppm. The correlation pattern revealed that the proton at H-1 of ADN, previously assigned to 5.85 ppm, exhibited correlations with the methine and methylene protons of β-CD at 3.32 and 5.70 ppm, respectively. The remaining signals observed in the ROESY corresponded to H-2 and the aromatic protons of ADN [42]. In the ^1^H NMR spectrum of the MOFs, the presence of aromatic and amino protons in the range of 8.12–8.35 ppm serves as evidence of the organic molecule ADN within the MOFs (Figure 9B). Assignments of these protons are developed based on the substituent and electronegativity effects of the groups present in ADN. Specifically, in the ^1^H NMR spectrum of the ADN:β-CD-K MOFs, the aromatic and amino protons are observed as singlets at 8.35, 8.33, and 8.12 ppm. These protons are conveniently assigned by comparing them with free ADN. Within the MOFs, all protons, excluding the hydroxy protons of ADN, displayed a deshielding effect, resulting in a shift ranging from 0.03 to 0.5 ppm. Conversely, the hydroxy protons exhibited a shielding effect, leading to a shift of 0.5 ppm (Appendix A). Therefore, the broad singlets observed at 4.95 and 4.75 ppm are assigned to the hydroxy protons at C-2′ and C-4′ of the ADN component, respectively. A multiplet attributed to the hydroxy methylene group is observed at 3.23 ppm, also showing a shielding effect of approximately 0.5–0.7 ppm. This chemical shift can be attributed to the interaction between the MOFs and ADN in the form of ICs. Additionally, in the MOFs, the hydroxy proton attached to the methylene carbon is exchanged with a potassium ion. For a detailed overview of the proton chemical shifts of ADN:β-CD-K MOFs, please refer to Appendix A. The proposed structure of ADN:β-CD-K MOFs is provided in Figure 9D.

### 3.3. In Vitro Release Profile of ADN from ADN:β-CD-K MOFs

The release of ADN from ADN:β-CD-K MOFs was investigated in this study. CDs-based ICs are capable of encapsulating ADN molecules within their structure. In this study, the release profiles of ADN from MOFs are investigated using a spectrophotometer in a buffer solution over a duration of approximately seven hours. To ensure consistent ADN content in the MOFs for the release study, the amount of ICs used was adjusted accordingly. ADN demonstrated a controlled release behavior, exhibiting an initial burst followed by a gradual and stable release profile, likely attributed to balanced conditions such as similar diffusion resistance at different time intervals. Three different pH conditions (acidic, basic, and neutral) are applied to investigate drug release. The results indicated that the neutral pH (6.5) provided the most favorable and efficient release condition (Figure 10). The release behavior of CDs on the surface of MOFs was comparatively lower under acidic and basic pH conditions, possibly due to the stable nature of ADN within the cavity of the CD. The formation of ICs facilitated the easier release of ADN from the MOFs, which is crucial for efficient drug delivery. Additionally, incorporating ADN into MOFs improved its dissolution in water, particularly when ICs are formed, further influencing the release behavior [61]. The release of ADN from the MOFs is directly linked to their biological activity. The sustained and controlled release of ADN can maintain therapeutic ADN levels within the desired range, leading to optimal efficacy. Consistent and prolonged drug release can ensure continuous exposure of ADN to the target cells or tissues, enhancing its biological activity [62].

### 3.4. In Vitro Cytotoxicity on Cell Lines

Cytotoxicity is performed on two different cancer cell lines to evaluate the effects of MOFs. Cell survival is measured. The experiments aimed to compare the cytotoxicity of the MOFs with the control. *In vitro* cell viability is performed using the Alamar blue assay on MRC-5 (normal cells) and MDA-MB-231 (cancer cells) cell lines [63]. The results, presented in Figure 11 and Appendix A, are expressed as mean ± SD (*n* = 3).

The Alamar blue assay results demonstrated a significant decrease in cell viability for both cell lines as the concentration of MOFs in the cells increased (Figure 11). These results indicate that there is a minimal difference in toxicity caused by MOFs between the two cell lines [64]. The normal MRC-5 cells exhibited lower toxicity compared to the cancer cells (MDA-MB-231). Assessing toxicity is crucial for investigating the safety of nanomaterials, and traditionally, toxicity analysis has involved animal studies. However, due to ethical considerations, many countries now regulate and limit animal testing. As a result, alternative methods for studying the toxicity of the derived materials are needed. The use of human cell models provides an effective, cost-efficient, and animal-free testing approach. In our investigation, the normal MRC-5 cells showed very low toxicity at a concentration of 2000 µg/mL (survival 85.65 ± 0.52%) (Figure 11) as observed in the bright field images (Figure 12), and they exhibited no toxic effects at other concentrations (Figure 10). On the other hand, cancer cells (MDA-MB-231) demonstrated toxicity at high concentrations of 2000 and 200 µg/mL (survival 72.28 ± 0.28% and 94.05 ± 0.61%, respectively) (Figure 11), while other concentrations did not induce toxic effects. Normal and cancer cells possess distinct characteristics, including cell propagation, cell division control, and apoptosis function [65]. Cancer cells rely on glucose metabolism for energy, and the MOFs inhibit glucose metabolism in these cells. Consequently, when assessing the cell viability of derived materials using cancer cells, the results differ significantly from those obtained with normal cells. Compared to normal cells, cancer cells inherently exhibit elevated levels of reactive oxygen species (ROS). The introduction of chemotherapy to cancer cells further elevates ROS levels. This oxidative stress disrupts the metabolism of cancer cells, specifically causing mitochondrial dysfunction, ultimately leading to apoptosis. In a 2020 study by Yang et al. [66], cancer cell lines such as HeLa, 4T1, and B16 showed mild injury when treated with the combination of 2-Deoxy-D-glucose (2DG) and doxorubicin (Dox), resulting in increased mitochondrial depolarization and ROS production. Importantly, normal cells are less affected by this treatment. These findings suggest that distinct metabolic and regulated pathways contribute to tumor cell death while minimizing damage to healthy tissues. The development of multidrug resistance (MDR) may involve mechanisms such as decreased drug efflux and the regulation of glucose metabolism through the ADN binding cassette transporter P-glycoprotein (P-gp) pathways.

The findings of this study demonstrate the potential cytotoxic properties of synthesized MOFs in targeting cancer cells, highlighting their promise in cancer treatment. The encapsulation and delivery capabilities of ADN from MOFs present a viable strategy for improving the efficacy of existing anticancer drugs and potentially overcoming drug resistance.

## 4. Conclusions

This investigation successfully achieves the loading of ADN into MOFs based on CD using a solvent diffusion method. Analysis of peaks associated with the ADN-loaded inclusion complexes (ICs) within the MOFs indicates that ADN becomes incorporated into a partially amorphous state. SEM images reveal well-defined crystalline structures within the MOFs. Interestingly, in the absence of ADN, smaller and irregularly shaped crystals are formed, potentially due to the MOF manufacturing process. The FT-IR spectral study explores the additional cross-linking of β-CD with K through the coupling of -OH on the β-CD-K MOFs, and confirms the crucial role of β-CD as a seed in the creation of β-CD-K MOFs. *In vitro* toxic effects of the MOFs are assessed by exposing them to MDA-MB-231 cells. Overall, this study successfully demonstrates the loading of ADN into CD-based MOFs, characterizes the structure and properties of the resulting MOFs, and investigates their potential applications in cytotoxicity assessment. However, further research and in vivo studies are required to validate the efficacy and safety of these CD complexes for potential clinical applications.

## Data Availability

The datasets used or analyzed during the current. Studies are available from the corresponding author upon reasonable request.

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
