# Peer review of "Adenosine/β-Cyclodextrin-Based Metal–Organic Frameworks as a Potential Material for Cancer Therapy"

_biomolecules, 2023, doi:10.3390/biom13071154_

Round 1

Reviewer 1 Report

Author provided sufficient details in introduction and performed experiments to support the anticancer efficiency of the complex ADN-beta-Cyclodextrin-MOF in normal and cancer cell lines. But lots of explanation need to complete this article. It seems ADN itself has more potent than ADN with CD-MOFs.

1. Author mentioned aromatic and amino protons chemical shift served as evidence of ADN within MOFs. But, there are no peaks in the spectrum.

2. How do author confirms the only one molecule of ADN encapsulated within the CD?

3. Did author try any 2D DOSY NMR spectra to confirm only one ADN molecule within CD?

4. In Fig (D), author used the arrows to draw CD structure. Please use the powerpoint to draw to avoid the arrows.

5. Author should show the 1H NMR spectra ADN with CD.

6. As per table S3, ADN itself shows better anticancer efficacy than ADN-beta-CD MOF. What is the purpose of this MOF?

7. Author should give reference to MOFs inhibit glucose metabolism in cells.

Reviewer 2 Report

1. Please check the reference citing format in the main text. It seems to be different.

2. When it deals with electrochemical, authors should better explain how MOFs have their unique advantages in these three aspects and provide some examples.

3. In the Fig.4, pls mark the scale in detail and clearly.

4. Pls Measure the BET data before and loading AND

5. Pls provide the releasing profile under different media. It is an important parameter

6. Some related refs could be updated, such as Dalton Trans, 2023, 52, 6226 – 6238; Theor. Chem. Acc. 2022, 141, 68. ACS Omega, 2018, 3, 17986−17990; Monatsh Chem, 2017, 48,1259–1267

7. The manuscript is not good writing. For example, The letter in Fig. 4 and 5 are larger than the main text unsuitably.

8. The authors should carefully polish their language.

revise

Round 2

Reviewer 1 Report

There are considerable improvements in the manuscript.

1. Please include the 1D NMR spectra of ADN:β-CD Inclusion complex  in 2D ROESY spectra in both X and Y axis.

2. Please mention all the NMR figure caption about NMR instrument MHz and  label the NMR solvent peaks.

Reviewer 2 Report

accept

Author Response

Thanks for your valuable comments.